# Gut bacteria are rarely shared by co-hospitalized premature infants, regardless of necrotizing enterocolitis development

Tali Raveh-Sadka[1], Brian C Thomas[1], Andrea Singh[1], Brian Firek[2], Brandon Brooks[1], Cindy J Castelle[1], Itai Sharon[1], Robyn Baker[3], Misty Good[3,4], Michael J Morowitz[2], Jillian F Banfield[1]*

[1]Department of Earth and Planetary Science, University of California, Berkeley, Berkeley, United States; [2]Department of Surgery, University of Pittsburgh School of Medicine, Pittsburgh, United States; [3]Division of Newborn Medicine, Children's Hospital of Pittsburgh and Magee-Womens Hospital of UPMC, Pittsburgh, United States; [4]Department of Pediatrics, University of Pittsburgh School of Medicine, Pittsburgh, United States

**Abstract** Premature infants are highly vulnerable to aberrant gastrointestinal tract colonization, a process that may lead to diseases like necrotizing enterocolitis. Thus, spread of potential pathogens among hospitalized infants is of great concern. Here, we reconstructed hundreds of high-quality genomes of microorganisms that colonized co-hospitalized premature infants, assessed their metabolic potential, and tracked them over time to evaluate bacterial strain dispersal among infants. We compared microbial communities in infants who did and did not develop necrotizing enterocolitis. Surprisingly, while potentially pathogenic bacteria of the same species colonized many infants, our genome-resolved analysis revealed that strains colonizing each baby were typically distinct. In particular, no strain was common to all infants who developed necrotizing enterocolitis. The paucity of shared gut colonizers suggests the existence of significant barriers to the spread of bacteria among infants. Importantly, we demonstrate that strain-resolved comprehensive community analysis can be accomplished on potentially medically relevant time scales.

*For correspondence: jbanfield@berkeley.edu

**Competing interests:** The authors declare that no competing interests exist.

## Introduction

Infection by potentially pathogenic and antibiotic-resistant bacterial strains is a major source of disease in hospitalized patients. However, the spread of bacteria among patients is hard to track because most methods cannot distinguish between closely related strains. Strain transmission is especially important during colonization of newborns, a process that is critical for proper development (*Arrieta et al., 2014*). Premature infants, in particular, are highly susceptible to aberrant colonization, as their microbiome is often disrupted by antibiotic treatments (*Greenwood et al., 2014*) and since the source of colonists likely includes the hospital environment (*Brooks et al., 2014*; *Taft et al., 2014*).

Necrotizing enterocolitis (NEC) is a common and life-threatening gastrointestinal disease that primarily affects hospitalized premature infants. Recent data indicate that ~7% of infants born weighing <1.5 kg develop NEC (*Neu and Walker, 2011*). Various observations support a microbial role in NEC, including the high incidence of pneumatosis intestinalis (gas in the bowel wall) in affected infants and resolution of symptoms in a majority of patients after antibiotic therapy and bowel rest

**eLife digest** The spread of potentially harmful bacteria is a major source of disease in patients staying in hospitals. Premature babies—born before 37 weeks of pregnancy—can be particularly vulnerable to these infections because their organs may not yet be fully developed. Also, young babies do not have the fully established populations of beneficial microbes that help to protect us from dangerous bacteria.

Necrotizing enterocolitis—a life-threatening disease that can cause portions of the bowel to die—is mostly seen in extremely premature babies. Although it is not known what causes this serious condition, research has suggested that a contagious microbe may be responsible.

The development of methods that can sequence DNA from whole communities of microbes, known as metagenomics, allows researchers to identify the presence of individual strains of bacteria within these communities. This makes it possible to compare and contrast the strains of bacteria present in both diseased and healthy individuals, to help identify the bacteria responsible for a disease.

Here, Raveh-Sadka et al. used a metagenomics approach to study the communities of microbes present in premature babies in a hospital unit during an outbreak of necrotizing enterocolitis. The study found that very few bacterial strains were present in more than one baby, suggesting that bacterial strains are not readily transferred between the babies while they are in the hospital. Furthermore, Raveh-Sadka et al. reveal that no single bacterial strain was shared among all the babies who developed necrotizing enterocolitis.

These findings indicate that necrotizing enterocolitis is not caused by a single strain of bacterium. Instead, if bacteria do contribute to the disease, it maybe that it is caused by a variety of potentially harmful bacteria colonizing the gut at the cost of beneficial bacteria. In future, better understanding of the barriers that limit the transfer of bacteria between premature babies could help inform efforts to reduce the spread of infections between patients in hospitals.

(*Grave et al., 2007*; *Morowitz et al., 2010*; *Carlisle and Morowitz, 2013*). NEC is characterized by intestinal inflammation and commonly progresses to necrosis, sepsis, and death. Risk factors may include feeding with artificial infant formula, blood transfusion, infant genetics, and overall health status (*Mally et al., 2006*; *Schnabl et al., 2008*; *Neu and Walker, 2011*; *Wan-Huen et al., 2013*). Such factors might be expected to give rise to a fairly constant disease incidence rate. However, NEC is commonly reported to occur in outbreaks (*Boccia et al., 2001*; *Meinzen-Derr et al., 2009*), suggesting involvement of a contagious microorganism. A review of 17 published outbreaks of NEC did not identify a reproducible pattern of bacterial infection (*Boccia et al., 2001*).

Cultivation-based approaches to identify and track medically relevant organisms can be labor intensive, biased, and inefficient. Yet, sequencing of the genomes of these cultured organisms can distinguish between strains with divergent phenotypes such as antibiotic susceptibility and virulence (*Didelot et al., 2012*), and has enabled analysis of pathogen dispersal (*Chin et al., 2011*; *Köser et al., 2012*; *Snitkin et al., 2012*; *He et al., 2013*). An alternative approach uses 16S rRNA gene sequencing to identify organisms without cultivation (*Brooks et al., 2014*; *Taft et al., 2014*), but the taxonomic resolution is limited, and distinct strains cannot be differentiated or tracked. Nonetheless, the method has been used to compare gut bacterial populations in fecal samples from infants with and without NEC. The results have been inconclusive. Some studies have identified no differences between cases and controls (*Normann et al., 2013*), while others have reported positive, but divergent findings (*Wang et al., 2009*; *Mshvildadze et al., 2010*; *Mai et al., 2011*; *Claud et al., 2013*; *Morrow et al., 2013*).

In contrast, whole community DNA sequencing methods (metagenomics) can profile microbial communities with strain resolution and probe the metabolic potential of community members (*Tyson et al., 2004*; *Gill et al., 2006*; *Kuczynski et al., 2012*). Applied to series of samples, the approach can document shifts in community structure and identify responses to medical treatments, increasing age, altered diet, and changing health status. Unlike 16S rRNA gene surveys, metagenomics does not rely on previously established information (e.g. conserved sequences that guide PCR-based rRNA-based detection) and is less likely to miss community members

(e.g. organisms with unusual rRNA sequences, phage, and plasmids). Compared to cultivation-based methods, the metagenomic approach provides a relatively unbiased view of community composition and thus may be particularly helpful when an unknown microorganism is the cause of a disease (*Relman, 2011*). A year ago, the power of such an approach was demonstrated in a retrospective analysis of banked samples from patients affected during a 2011 outbreak of a severe diarrheal illness caused by Shiga-toxigenic *Escherichia coli* (*Loman et al., 2013*). More recently, shotgun sequencing of bacterial DNA present within cerebrospinal fluid enabled the diagnosis and treatment of leptospirosis in a critically ill child with meningitis (*Wilson et al., 2014*).

Among gastrointestinal diseases with a possible microbial origin, NEC is somewhat unique as it is relatively common and because samples that provide information about gut consortia can be collected prior to the development of symptoms. This is because infants at risk for NEC are typically hospitalized for weeks to months in the neonatal intensive care unit (NICU) and onset of the disease occurs over a defined time period. Only one small study that we are aware of has analyzed metagenomic sequence data from infants with and without NEC (*Claud et al., 2013*), but assembly of the sequences was not attempted.

Recently, a group of infants developed NEC over a short time period in the NICU of Magee-Womens Hospital of the University of Pittsburgh Medical Center. Here, we investigated the degree to which specific microbial strains were shared among co-hospitalized infants and whether the disease could be attributed to a single infectious agent. Because the analysis required confirmation that the same strain was present in multiple infants, we deployed a genome-resolved sequencing-based approach. Our analyses included consideration of the fastest evolving features of genomes (e.g. prophage and the CRISPR/Cas loci) to maximize strain resolution. We also investigated strain-level metabolic potential and evaluated population heterogeneity for one abundant and widespread species. Genome-resolved approaches are typically slow and bioinformatics intensive because the data sizes are massive, simultaneous reconstruction of genomes for multiple community members is complex, and comparative and metabolic analyses for hundreds of genomes are challenging. We applied a new analysis system to resolve data into genomes and analyze the metabolic potential. To the best of our knowledge, this study is the first to provide comprehensive, genome-resolved analysis of gut bacterial communities in co-hospitalized patients. The core methods are fast enough to make them useful in some clinical settings, and we anticipate that analysis time can be substantially decreased with future developments.

## Results and discussion

### Clinical information regarding study subjects and the NEC cluster

When it became apparent that the incidence of NEC was increasing, we selected five infants who had developed NEC and five controls for comprehensive microbial community analysis. Ultimately, during the summer of 2014, nine infants were diagnosed with NEC (Bell's stage II or III). The total number of NEC cases was 10, as one of the affected infants developed recurrent NEC. This incidence rate was 2.5 times higher than average for this NICU.

For affected infants #2 (who developed NEC twice), #3, and #8, multiple fecal samples collected prior to the onset of symptoms were available. Two additional infants, #9 (not premature) and #10, were enrolled after diagnosis and treatment. The other infants who developed NEC were not enrolled in our study. Four infants (#1, #4, #6, and #7) did not develop NEC. Infant #5 was not diagnosed with NEC but had a single bloody stool on day of life (DOL) 20 and was treated with antibiotics for a suspected urinary tract infection. All infants were hospitalized concurrently within the same NICU (i.e. synchronous controls), and several were matched also according to gestational age. The selection of samples for sequencing was aimed to provide dense sampling around diagnosed NEC cases, from both the diagnosed infants as well as co-hospitalized infants who did not develop NEC (see sampling schedule in *Figure 1* and additional medical details in *Supplementary file 1*). Bacterial load in each sample was quantified by ddPCR (see 'Materials and methods' section). The estimated load was in general agreement with previous measurements in full-term infants of similar postnatal ages (*De Leoz et al., 2014*). Notably, the variation in the number of microbes per gram feces did not exceed a 100-fold across all samples. Infants who developed NEC did not show a consistent trend of change in bacterial load prior to or following diagnosis (*Figure 1*).

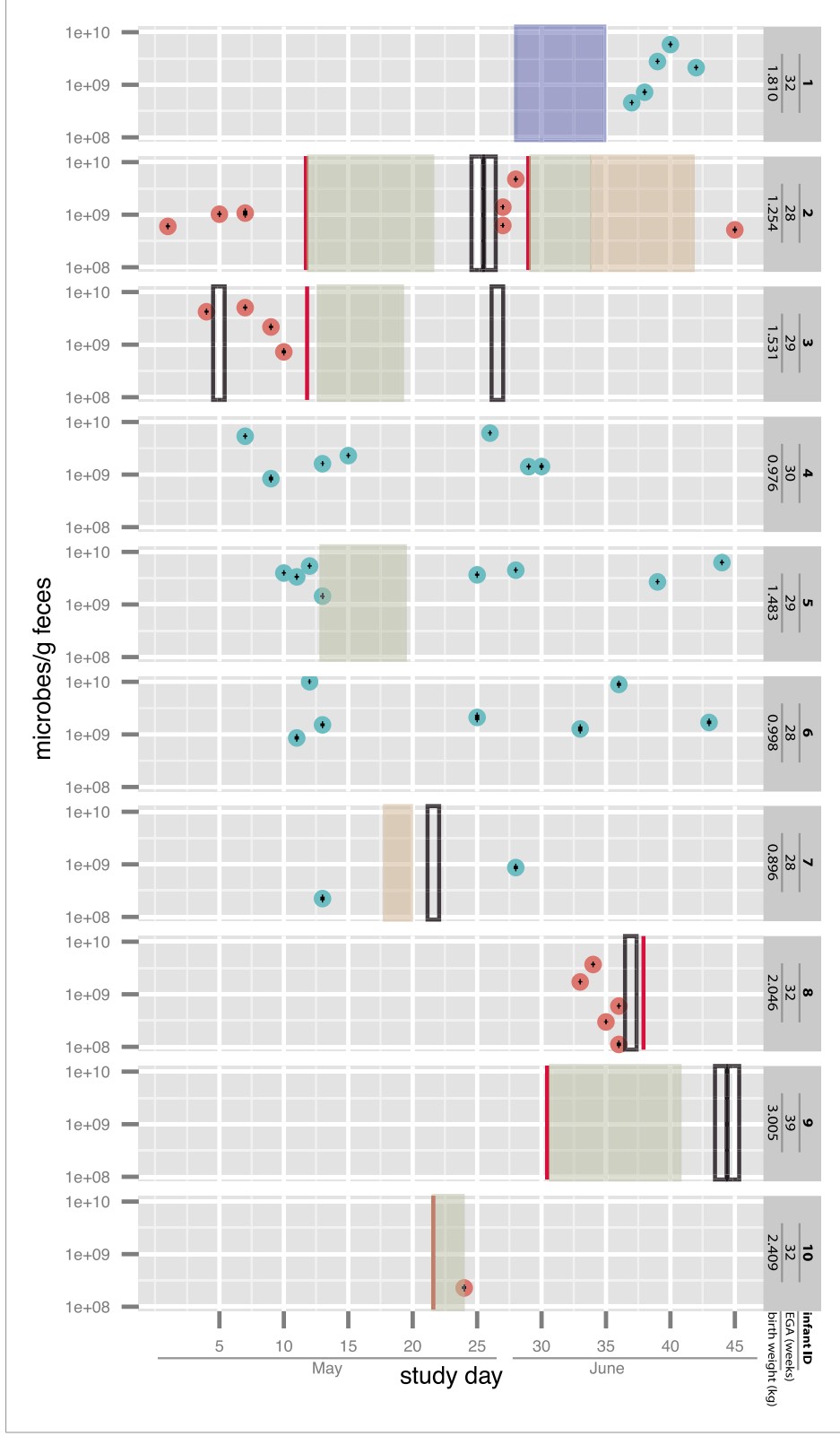

**Figure 1**. Overview of the sampling of infants affected by necrotizing enterocolitis (red) and controls (blue) and microbial cell loads based on droplet digital PCR (ddPCR) quantification of fecal samples. For ddPCR, standard deviations for triplicates are plotted within each data point. Also shown are necrotizing enterocolitis (NEC) diagnosis

*Figure 1. continued on next page*

*Figure 1. Continued*

times (vertical red lines) and periods of antibiotic administration: green: ampicillin + cefotaxime, orange: vancomycin + cefotaxime, and blue: ampicillin + gentamycin (see *Supplementary file 1*). Black boxes indicate metagenomic samples for which insufficient sample remained for ddPCR. EGA: estimated gestational age.

## Assembly and rapid binning yields hundreds of near-complete genomes

DNA was extracted from up to nine samples per infant and sequenced using an Illumina HiSeq2500 at the University of Illinois. Overall, we analyzed 55 samples from the 10 infants (*Figure 1* and *Supplementary file 2*). Between 2.22 and 7.35 Gbp of trimmed data from each sample was assembled independently. This enabled at least 4× coverage for genomes of organisms that comprised more than ~0.2% to ~0.6% of each community. For the 10 datasets, we assembled 181.2 Gbp of read sequence information. In total, 1.35 Gbp of genome sequence was generated on scaffolds >1000 bp (see 'Materials and methods' section and *Supplementary files 2 and 3*).

The genome reconstruction strategy involved a user assigning scaffolds to organisms using online binning tools (see 'Materials and methods' section). Genome bins were defined based on a combination of a phylogenetic profile, GC content, and coverage. These bins were then verified independently using emergent self organizing maps (ESOMs) that clustered either tetranucleotide composition or time series abundance pattern information. Genome completeness and purity were evaluated based on the inventory of ribosomal proteins and 51 genes expected to be in single copy in any genome (see 'Materials and methods' section for details).

A total of 509 bacterial genomes (including multiple genomes for the same organism in different samples) were recovered, with average read coverage of between 2 and 1148. Between 1 and 23 bacterial genome bins were detected per sample, and overall 260 near-complete genomes were reconstructed (see 'Materials and methods' section). Scaffolds identified as putative phage or plasmids based on their encoded genes were assigned to 328 bins (*Supplementary file 3*). Overall, between 86% and 98% of reads generated for each sample was assigned to a genome bin (*Supplementary file 2*).

## Diverse bacterial strains in the NICU are rarely shared by co-hospitalized babies

In order to assess the extent of strain dispersal among the hospitalized infants, genome bins with >0.5 Mbp of sequence were compared by aligning the single copy genes sequences. When these were too fragmented for conclusive results, entire genome bins were aligned. Genome bins that were >98% identical across >90% of bin length were considered indistinguishable. Manual curation of assemblies was performed in some cases to eliminate disagreements due to scaffolding errors that are introduced occasionally during assembly (see 'Materials and methods' section).

Remarkably, very few bacterial strains occurred in more than one infant and no strain was shared by all infants who developed NEC (*Figure 2*). In contrast, and as could be expected, identical genotypes were almost always detected in samples from the same infant, providing reassurance regarding the validity of our methods (*Figure 2*).

Specifically, of the 149 strains compared, only four were shared by two or more infants, and only three of these were identified in infants who developed NEC. A *Klebsiella oxytoca* strain was present in infants #1 and #6, neither of whom developed NEC. *Clostridium sporogenes* was present in infants #3 and #5, but occurred at very low abundance in infant #3. Two strains were more widely distributed: a *Clostridium butyricum* strain was detected in infants who did (infants #3, #8) and did not (infants #1, #5, #6) develop NEC but was missing from infant #2, who developed NEC. *C. butyricum* has no predicted type III or type VI secretion system genes and no identified toxin-producing genes (*Supplementary file 4*). Thus, this strain seems unlikely to be a pathogen or the cause of NEC. Lastly, a *Clostridium paraputrificum* strain with a moderate predicted pathogenicity potential (*Supplementary file 4*) was shared by infants #2, #5, and #8, and also occurred in one sample from infant #6, although the predominant strain in this infant (who did not develop NEC) was different (*Figure 2*). *C. paraputrificum* was not detected in infant #3 (NEC case). Both *C. paraputrificum* and *C. butyricum* have been previously suggested as potential causative agents in NEC (*Waligora-Dupriet et al., 2005*).

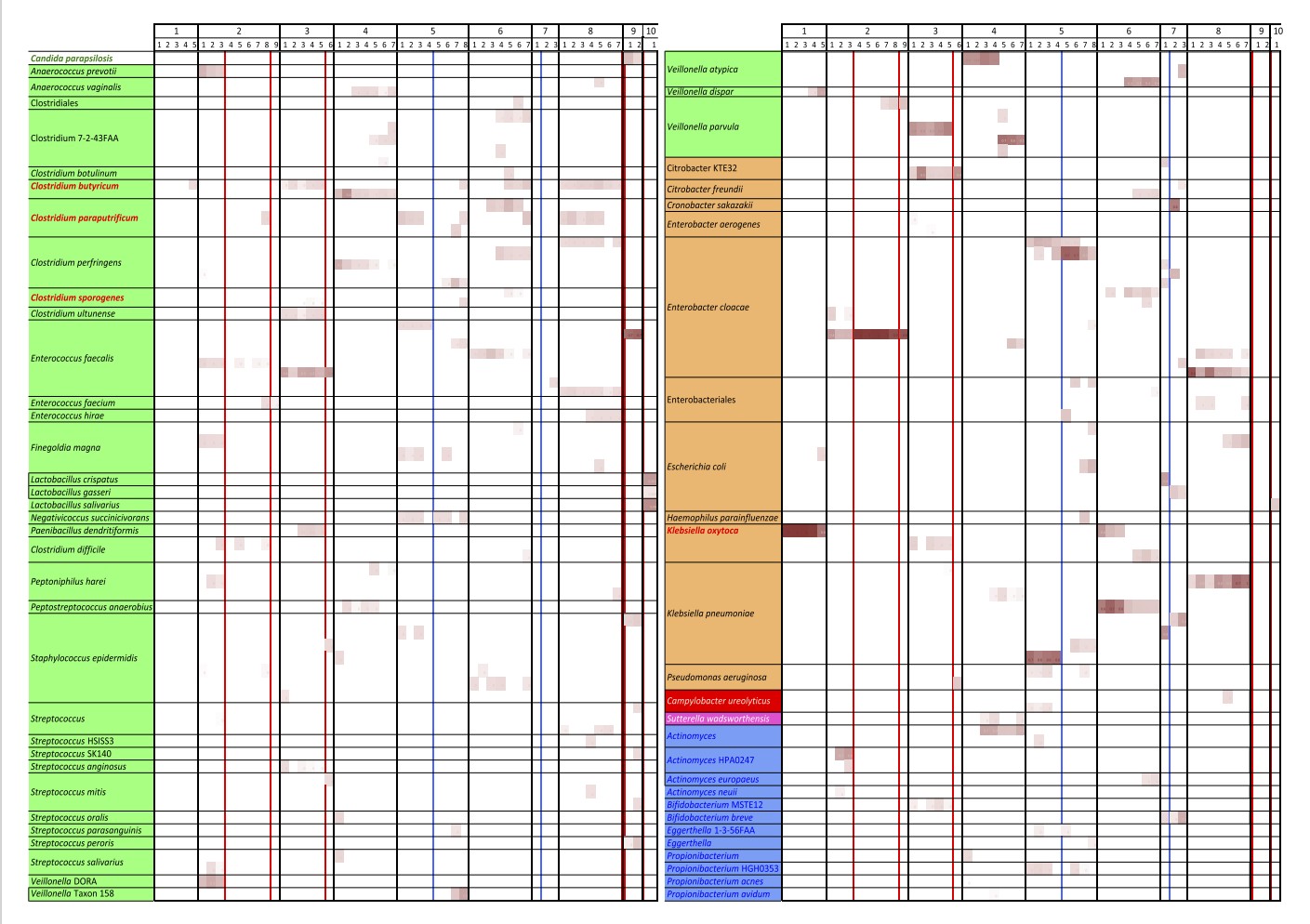

**Figure 2**. An overview of the distribution of 144 of the 149 tracked strains in the 55 samples from 10 infants (five rare organisms were not included for space reasons). White boxes indicate that the strain was absent; shading intensity increases with increased organism abundance. Note the persistence of specific genotypes within infants and the almost complete lack of overlap in strains between infants. The few strains shared between infants are highlighted in red. Colors associated with organism names indicate the broader organism classification: green are Firmicutes, orange are Gammaproteobacteria, red are Epsilonproteobacteria, pink are Betaproteobacteria, and blue are Actinobacteria. Red lines indicate antibiotic administration associated with necrotizing enterocolitis diagnoses, blue lines indicate antibiotic administration for other reasons.

Interestingly, although the colonizing strains were almost always distinct, infants often shared bacteria of the same genus or species. At high abundance in multiple infants, including two who developed NEC, were members of the genus *Veillonella*. However, multiple distinct strains and species were present across infants (*Figure 3*). A *Veillonella* strain was very abundant in infant #2 prior to development of NEC (*Supplementary file 3*) but disappeared after the first antibiotic treatment, to be replaced by different *Veillonella* species (*Figure 2*). In the other infants who developed NEC, *Veillonella* was either absent (infant #8) or present as a different strain altogether (infant #3). The results likely rule out a *Veillonella* strain as a single, shared agent of NEC.

Another organism found in multiple infants, often at high abundance (*Figure 2* and *Supplementary file 3*), was *Enterococcus faecalis*. This organism is common in fecal samples from both premature and term infants (*Chang et al., 2011*; *Costello et al., 2013*; *Vallès et al., 2014*). Interestingly, it appears that the host–*E. faecalis* relationship, as it pertains to the infant gut, is nuanced. Although *E. faecalis* has repeatedly been identified as a source of neonatal infection (*Stoll et al., 2002*; *Härtel et al., 2012*), it also has been studied as a potential probiotic (*Nueno-Palop and Narbad, 2011*), with beneficial properties related to modulation of innate

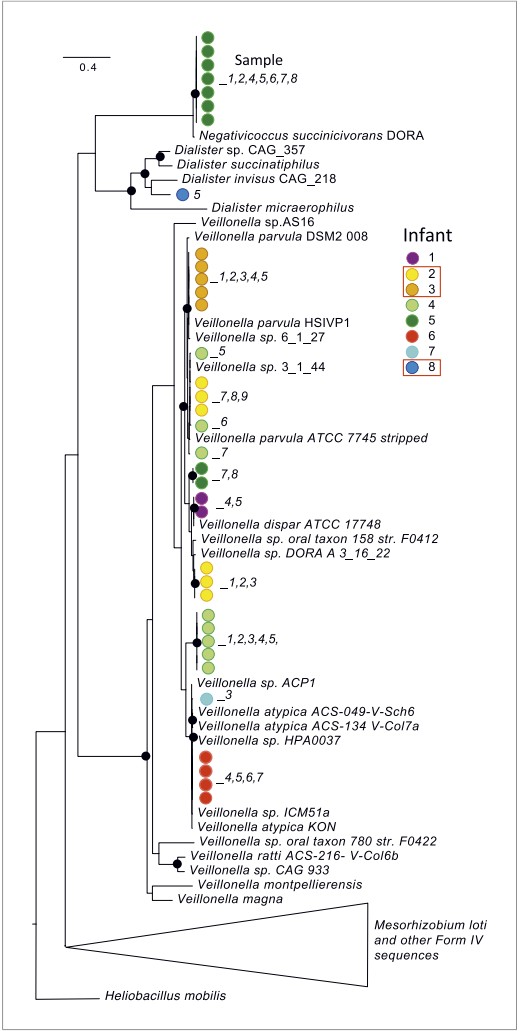

**Figure 3**. A phylogenetic tree (RAXML; black dots indicate bootstrap values of ≥80%) for predicted RuBisCO Form IV (RuBisCO-like) proteins involved in methionine salvage. This protein was chosen for analysis because it is well studied and is not one of the 51 single copy (and generally highly conserved) genes used in other analyses. Colored dots identify the infant, while the number indicates the sample of origin. Red boxes highlight infants who developed necrotizing enterocolitis (NEC). Although *Veillonella* were prominent in many samples, sequence analysis revealed many distinct strains/species over the study cohort. Strain shifts occurred following antibiotic administration (e.g. in infant #2), but identical sequences were often detected in series of samples from the same infant. Note infants affected by NEC do not share the same strains/species.

immunity (*Wang et al., 2014*). Furthermore, links between mobile genomic elements and enterococcal virulence are well described (*Gilmore et al., 2013*). These considerations suggest that strain-level variation in *E. faecalis* is significant and potentially clinically relevant.

For *E. faecalis*, we reconstructed 30 near-complete genomes (*Supplementary file 3*) for multiple strains (*Figure 2*). Alignment of the longest of the single copy genes tracked, the ~2500 bp DNA gyrase subunit A (*gyrA*) gene, illustrates five distinct sequence types for this gene alone (*Figure 4A*). Strains recovered from infants #2 and #7 and also strains recovered from infants #3 and #5 (early samples) could not be distinguished by this locus. Notably, reconstructed 16S rRNA gene sequences were identical in these strains, illustrating that the limited taxonomic resolution of this locus prevents its use in studies of strain dispersal.

A genomic region of interest for strain-level studies is the CRISPR/Cas locus. This locus can be one of the fastest evolving regions of bacterial genomes and thus can potentially provide high-resolution insight into strain distinction, as well as shared ancestry (*Tyson and Banfield, 2008*). All 30 well-sampled *E. faecalis* genomes encode a CRISPR spacer-repeat array that lacks proximal Cas proteins and some genomes (in infants #3, #5, #8) encode an additional locus with proximal Cas genes (*Figure 4B,C*). Given that Cas proteins are required for CRISPR-Cas function, strains in infants #2, #6, #7, and #9 that lack Cas proteins altogether, have lost CRISPR-Cas-based phage immunity.

This pattern of loci with and without Cas proteins has been reported previously in *E. faecalis* (*Palmer and Gilmore, 2010*). Different Cas1 sequences (types a and b) and a different repeat sequence were identified in *E. faecalis* from infants #3, #5 before antibiotic treatment, and #8, compared to the strain in infant #5 after antibiotic treatment (*Figure 4B*). The repeat-spacer arrays in the loci with Cas1 type a are identical in the genotypes of *E. faecalis* in infant #3 and in early samples from infant #5 (*Figure 5A*), reinforcing the very high similarity of these populations deduced from single copy gene sequence comparisons (*Figure 2*). As often happens in CRISPR loci (*Tyson and Banfield, 2008*), a block comprising six spacers and flanking repeats has been excised in the strain from infant #8 and three novel spacers have been added at the growing tip, versus two in infants #3 and #5 (*Figure 5A*). Shared spacers at the older end (distant from the Cas) imply that the strains in infants #3, #5, and #8 had a recent common ancestor.

The Cas-less CRISPR array is flanked by DNA-related and antibiotic resistance-related genes, and differs in length considerably among strains. The repeat for the locus without Cas proteins is identical

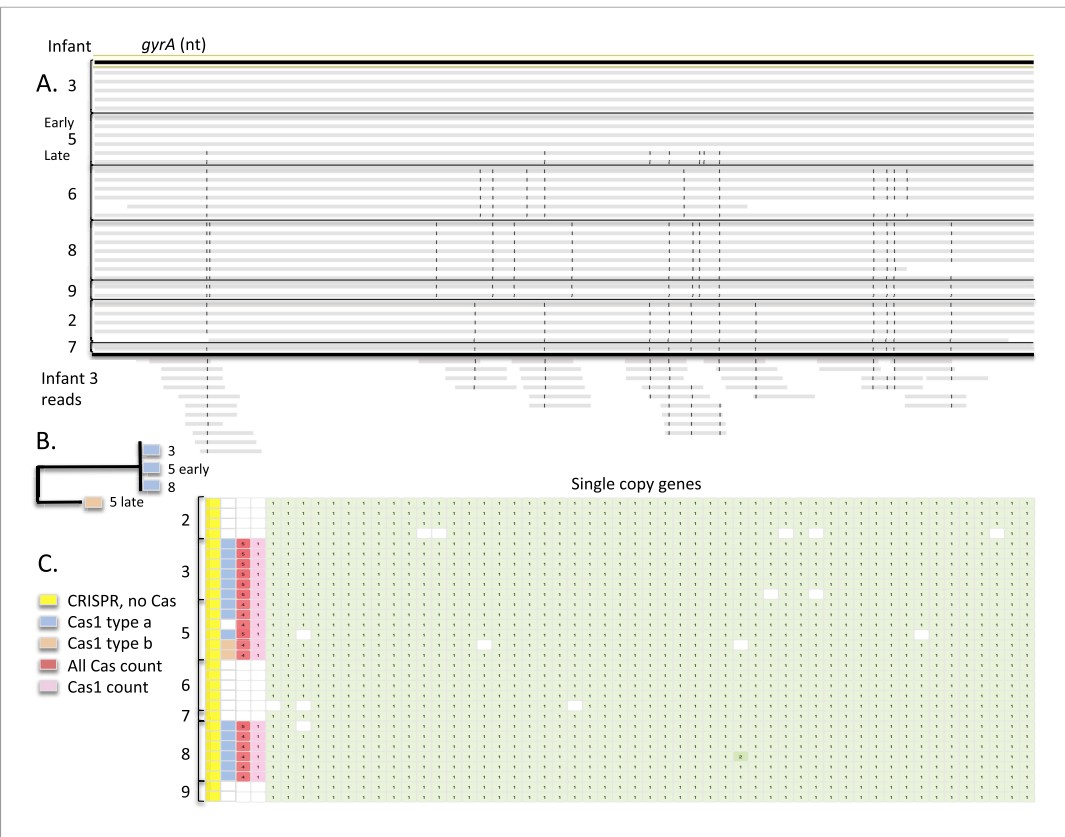

**Figure 4**. Strain differences in recovered *Enterococcus faecalis* genomes. (**A**) Alignment of the ~2500 *Enterococcus faecalis gyrA* nucleotide sequences from all infants to that from infant #3, sample 1 revealing five distinct types (gray bars are scaffolds; SNPs are vertical black lines). Shown below are a tiny subset of reads from infant #3, sample 4 with SNPs that match nucleotides in the *gyrA* sequences from *E. faecalis* in another infant; all SNPs are consistent with a strain very similar to that in infants #2 and #7 (although derivation of some reads from other strains cannot be ruled out). (**B**) Phylogenetic representation illustrating two distinct Cas1 sequence types. (**C**) Inventory of 51 single copy genes showing that the 30 *E. faecalis* genomes are near-complete and providing information about encoded CRISPR and Cas.

The following figure supplements are available for figure 4:

**Figure supplement 1**. Alignments showing single nucleotide polymorphisms (vertical colored lines on gray bars that represent the sequences) in the Histidyl-tRNA synthetase genes that distinguish from *Enterobacter cloacae* strains across samples and infants.

**Figure supplement 2**. Aspartyl-tRNA synthetase from *Klebsiella pneumoniae* strains in samples from infants #4, #5, #6, #7, and #8.

to that of the loci with type a Cas1. All first repeats are defective, but the polymorphisms are only shared by strains in infants #3, #5, and #8. However, the repeat-spacer array distinguishes the genotype in infants #3 and #5 from that in infant #8 (*Figure 5B*). The loci in the strains in infants #2 and #7 are probably the same (the sequence from infant #7 is not shown due to very partial recovery).

Both the single copy gene and CRISPR-Cas analysis suggested that *E. faecalis* in infants #2 and #7 are very closely related. Similarly, the strains in infant #3 and in early samples from infant #5 are almost identical (a single SNP in the surveyed gene set distinguished the sequences). To gain better understanding of the type and extent of genomic differences between the recovered *E. faecalis* genomes, and specifically of these closely related genome pairs, we mapped reads from eight samples, representative of the eight different genotypes reported in *Figure 2*, to a 1 Mbp *E. faecalis* scaffold recovered from infant #9, sample 1 (one third of the recovered genome). Multiple alignment of the consensus sequences from mapping of each sample provided a view of sequence variability

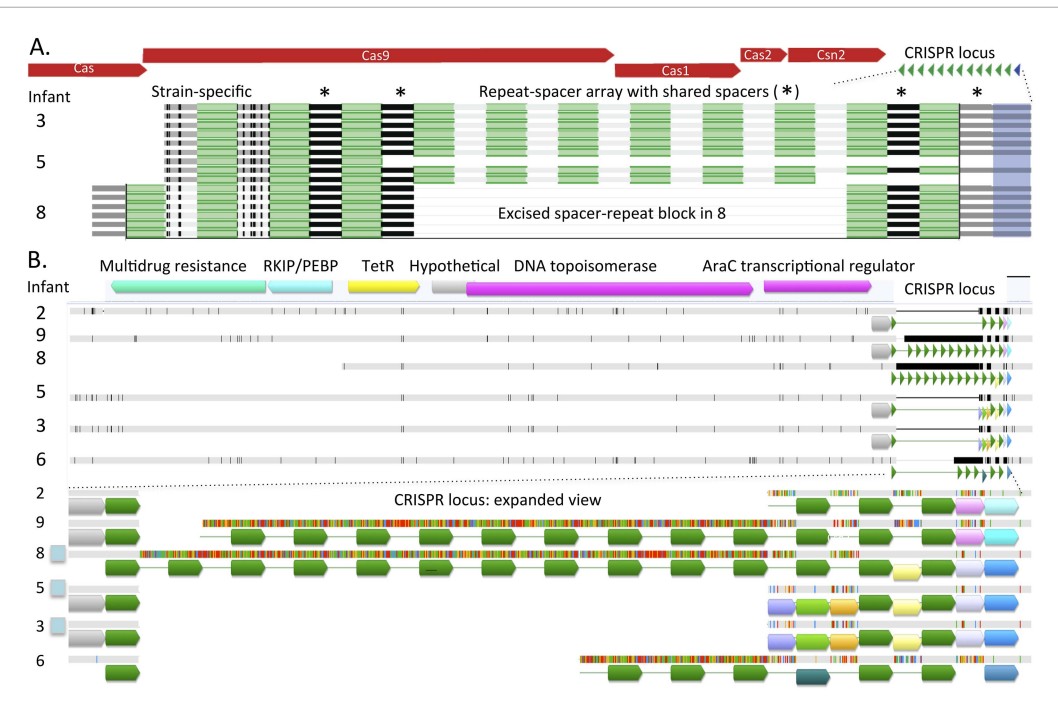

**Figure 5**. Comparison of CRISPR loci in *Enterococcus faecalis* genomes. (**A**) The CRISPR-Cas loci in infants #3, #5 (early strain), and #8 and (**B**) the CRISPR locus lacking adjacent Cas proteins. The first defective repeats are shown in blue, other repeats are in green. The CRISPR loci are expanded below. In **A**, two versus three spacers have been added to the young end of the loci (left side, adjacent to Cas) in infants #3, #5 versus #8, respectively. In **B**, scaffolds encoding the loci are shown as horizontal gray bars (polymorphisms in the multi-sequence alignment are small vertical tic marks). The same color indicates shared sequences. Blue boxes to the left indicate that the genome encodes Cas proteins. Both loci (**A** and **B**) are identical in infants #3 and #5.

across strains (*Figure 6A*; a similar alignment for *C. paraputrificum* strains is shown in *Figure 6—figure supplement 1*). The analysis revealed many SNP locations and small indels that were spread across the entire length of the sequence, as well as a small number of longer (20–30 Kbp) indel regions. These regions included among other things a sucrose metabolism operon, mobile elements, and genes related to Fe-S protein biogenesis.

The high degree of similarity between the closely related genotypes (infants #2 and #7, infants #3, and early samples of infant #5) is evident from inspection of the multiple alignment (*Figure 6B*). While, on average, sequence pairs were ~90% identical across the scaffold length, those of infants #2 and #7 and those of infants #3 and #5 (early samples) showed identity of 99.3% and 98.7%, respectively. However, a more comprehensive comparison of these genome pairs revealed differences in the genotypes in both cases. Interestingly, a ~1 Kbp region distinguishes strains in infants #3 and #5 (*Figure 6C*), whereas a prophage insertion separates those in infants #2 and #7 (*Figure 6D*). Both of these regions were missing from the genome recovered from infant #9, sample 1 and thus could not be discovered by read mapping to that genome. These differences are very subtle, and especially in the case of the prophage, might have arisen after colonization.

While the above analysis characterized differences in the major strains detected in the hospitalized infants, we further investigated whether *E. faecalis* strains from one infant occur at low or even trace levels in other infants. This was done by analysis of sequence polymorphisms in reads that map to the *gyrA* gene in each assembly. While in all infants colonized by *E. faecalis*, some polymorphic locations could be identified, in most infants, polymorphisms matching strains of other infants were undetectable (infants #2, #6, #8, #9). For the abundant population in sample 2 from infant #9, analysis of >20,000 reads indicated that the maximum abundance level of a strain detected in another infant was <0.01% (*Supplementary file 5*). However, in infant #3, ~0.12% of reads have sequences consistent with derivation from the genotype in infants #2 and #7, which are indistinguishable at this

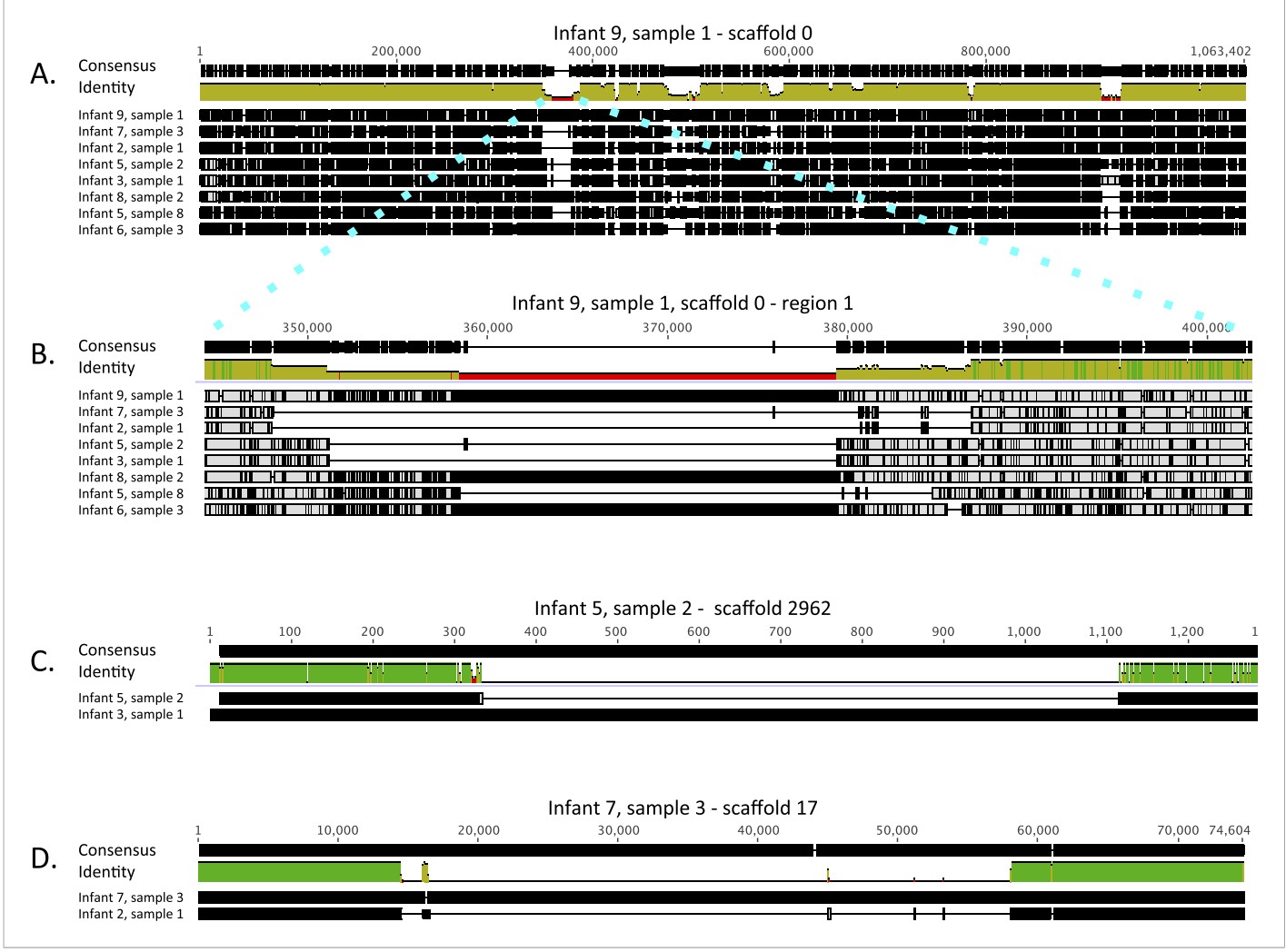

**Figure 6**. Alignment view of genome-wide differences in *Enterococcus faecalis* strains. Consensus sequence for the alignments (shown at the top of each alignment) represents the calculated order of the most frequent nucleotide residues. Alignments were done in Geneious v7.1.7 (***Kearse et al., 2012***), using MAFFT v7.017 (***Katoh et al., 2002***) with default parameters. Samples are ordered by similarity. For each sample, SNPs and indel locations relative to the multiple alignment are marked by black lines or boxes. (**A**) Reads from eight samples, from which different *Enterococcus faecalis* strains were recovered, were mapped to a 1 Mbp *E. faecalis* scaffold (scaffold 0) recovered from infant #9, sample 1. Shown is a multiple alignment of the consensus sequences derived for each sample from these mappings. Multiple SNPs and short indels are detected throughout the sequence. Several larger indels are also detected. (**B**) Enlarged view of a region in **A** showing a large indel locus. This view distinguishes sets of extremely closely related strains (i.e. strains in infants #7 and #2; strains in infants #3 and #5 [early samples]) from more distant strains. (**C**) Pairwise alignment of consensus sequences derived from read mapping to an *E. faecalis* scaffold (scaffold 2962) recovered from infant #5, sample 2 distinguishes closely related strains in infants #3 and #5 (early samples). (**D**) Pairwise alignment of consensus sequences derived from read mapping to an *E. faecalis* scaffold (scaffold 17) recovered from infant #7, sample 3 distinguishes closely related strains in infants #2 and #7. The region missing in the assembly from the other infants corresponds to a mobile element.

The following figure supplement is available for figure 6:

**Figure supplement 1**. Alignment view of genome-wide differences in *Clostridium paraputrificum* strains.

locus (***Figure 4A***). The *gyrA* analysis indicated that this genome is even more prominent in early-collected samples from infant #5 (3–9% of reads; ***Supplementary file 5***). Thus, compared to other *E. faecalis* strains, the population in infants #2 and #7 is relatively widely distributed.

The evidence from all infants is that multiple strains and species are present in the NICU. The intriguing pattern of mostly infant-specific *E. faecalis* could have arisen due to a very small number of colonizing *E. faecalis* cells. However, investigation of population-level sequence variation revealed

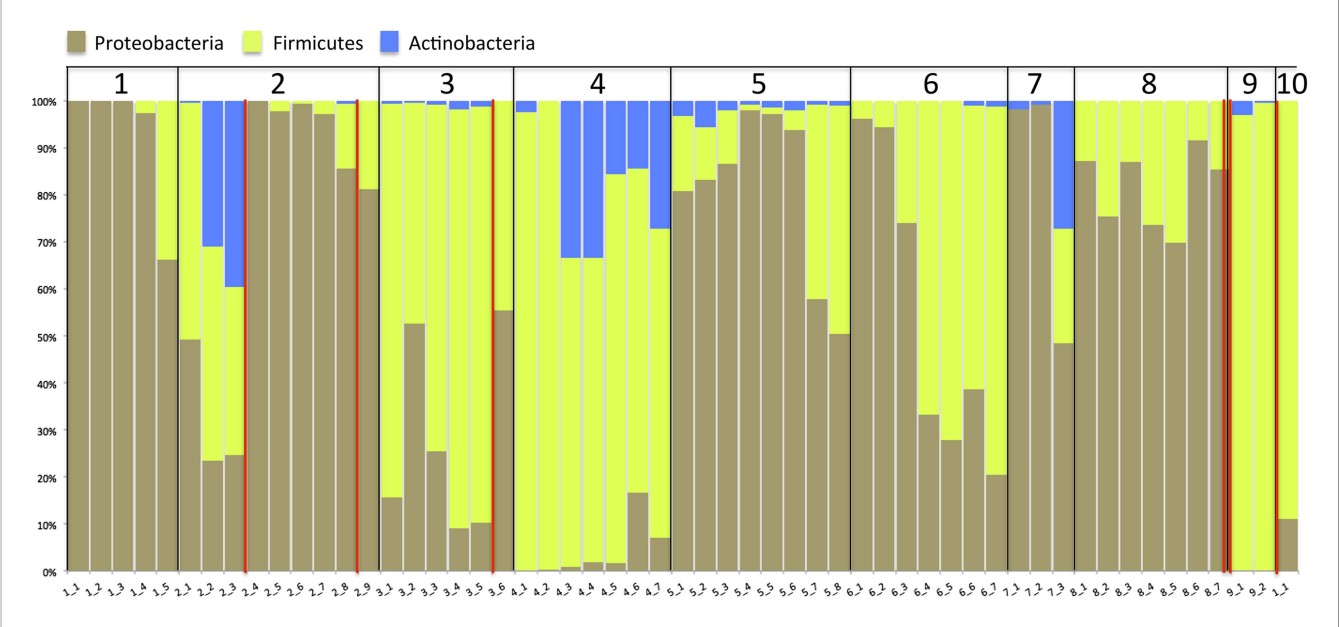

**Figure 7**. Stacked bar plot of community composition across samples and infants after organism identifications were collapsed to the phylum level to allow comparison to prior studies. Red lines indicate necrotizing enterocolitis diagnoses.

some reads with shared polymorphisms that are not characteristic of the strains in the other infants. This, and the detection of the dominant strain from infants #2 and #7 in other infants indicate that multiple *E. faecalis* inoculation events occur during colonization.

The lack of overlap in strain genotype could indicate the existence of infant-specific strain sources (e.g. mothers), and barriers that prevent spread of those populations to other infants in the NICU. Even if dispersal occurs, the founding population may preclude establishment of later introduced populations. Alternatively, strains could be dispersing freely, and strain dominance could reflect strong selection in the gastrointestinal tract (possibly imposed by microbial community context and/or human genetics) leading to the establishment of a single, most adapted strain. Another model worth considering would involve stochastic acquisition of a strain from a set of populations that is so large that it is improbable that any two infants would initially acquire the same strain. As there is ongoing input of strains over the colonization period, the observation of one (usually) highly dominant population still suggests some barrier to establishment of other populations.

## Phylum-level community composition does not distinguish NEC cases from other infants

Previous studies that were done at lower resolution than achieved in the current study have pointed to a high abundance of Proteobacteria as a factor in NEC development (*Wang et al., 2009*; *Mai et al., 2011*; *Torrazza et al., 2013*). To see if that pattern applied in the current study, we collapsed our organism identifications to the phylum level (*Figure 7*). Proteobacteria are abundant in most infants, but Proteobacteria representation in the communities did not distinguish those infants who developed NEC from those that did not. In fact, the relative abundance of Proteobacteria declines in infant #2 prior to both NEC diagnoses. Abundances are generally low in infant #3, and consistently high in infant #8.

## Community functions do not distinguish infants who developed NEC from other infants, but reveal many potential pathogens

Given that no single organism could be associated with all NEC cases, we considered the possibility that overall metabolic imbalance was a contributing factor. Owing to the high quality and completeness of many of the recovered genomes, we are able with this type of data, to go beyond strain identification and search for unique characteristics in the predicted gene content of colonizers of

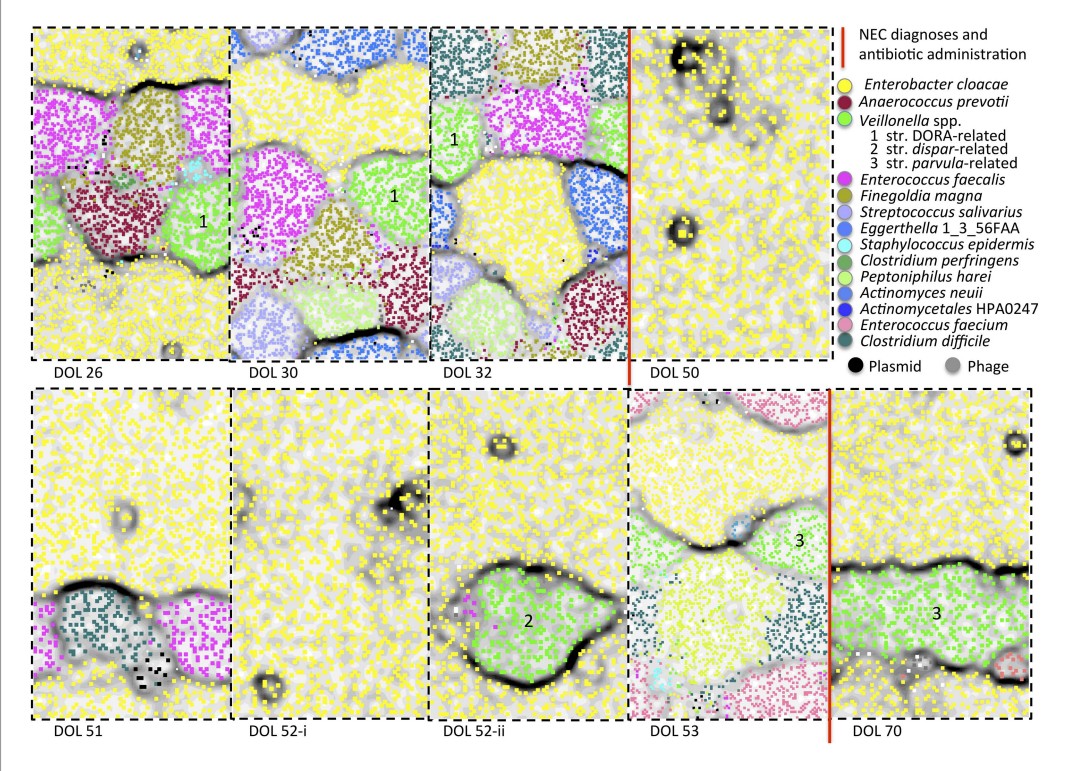

**Figure 8**. Microbial community composition, community complexity, and an overview of binning for samples from infant #2. The diagrams are unit repeats of a tetranucleotide emergent self organizing map; points coded to reflect the bin assignment of the scaffold verify the binning (see 'Materials and methods' section). Vertical red lines separate samples before and after antibiotic administration to treat necrotizing enterocolitis (NEC) (two instances). Organisms are listed primarily in order of abundance in the first sample. Note that, with the exception of the dominant member, *Enterobacter cloacae*, species representation changed dramatically following antibiotic administration. The *Veillonella* strain varied (numbers differentiate areas that represent different populations).

The following figure supplements are available for figure 8:

**Figure supplement 1**. Rank abundance curves describing the microbial community (exclusive of phage and plasmids) in infant #2.

**Figure supplement 2**. An overview of the microbial communities from infant #3.

**Figure supplement 3**. Overview community composition for infant #8, who developed necrotizing enterocolitis 1 day after collection of the last sample.

NEC-affected infants. However, clustering analyses of the detailed profiles of genomically encoded metabolic capacities (*Supplementary file 4*) and inspection of individual patterns failed to distinguish the capacities of microbial communities in infants who did and did not develop NEC. Of course, other considerations, such as differences in gene expression levels, may play an important role in NEC, but cannot be studied with DNA sequence information.

While no metabolic imbalance was found, many pathogens were detected in fecal samples of infants who developed NEC. For example, *Enterobacter cloacae* is abundant in infant #2 and is predicted to have many toxin and type VI secretion system genes and an extensive antibiotic resistance repertoire (*Supplementary file 4*). The resistance genes may explain why *E. cloacae* remained abundant after antibiotic treatment (sequence analyses indicate that the same genotype persisted through the treatments; *Figure 2* and *Figure 4—figure supplement 1*). ESOMs used to validate the binning also provide an overview of the community composition, and in the case of infant #2, also highlight the almost complete dominance of *E. cloacae* in response to antibiotics (*Figure 8*).

Note that these maps do not reflect organism abundances, although very small areas can indicate genomes that were partially sampled due to low sequence coverage (for abundance information see *Supplementary file 3* and *Figure 8—figure supplement 1*).

Many other potential pathogens are present prior to the NEC diagnoses in infant #2 (see *Figure 8* and *Supplementary file 4*). Of interest due to their predicted gene complement are an *Actinomyces* strain that dominated the community prior to the first event and *Clostridium difficile*, which occurs in both samples (at low abundance) collected immediately prior to onset of NEC. *C. difficile* has been implicated as a cause of NEC (*Han et al., 1983*; *Pérez-González et al., 1996*), although its role is controversial (*Boccia et al., 2001*; *Mshvildadze et al., 2010*). Notably, the genome of the organism in infant #2 encodes *tcdABCDE* genes characteristic of toxigenic strains (*Dingle et al., 2014*) and Clostridial binary toxin B/anthrax toxin PA family proteins are affiliated with this organism. Also of potential significance in infant #2, from the perspective of its genetic repertoire, were *Clostridium perfringens*, *Streptococcus salivarius*, *Enterococcus faecium*, and *E. faecalis* (*Supplementary file 4*).

Prominent in the communities of infant #3 prior to NEC diagnosis were *Veillonella parvula* (a strain predicted to have minimal pathogenicity, see *Supplementary file 4*, and unique to this infant; *Figure 3*), *E. faecalis*, *K. oxytoca*, and a *Citrobacter* related to strain KTE32 (*Figure 8—figure supplement 2*). The same strains of *E. faecalis* and *Citrobacter* persist through treatment, likely reflecting their large repertoire of antibiotic resistance genes (*Supplementary file 4*). *Citrobacter* and *Klebsiella* have many toxin and type VI secretion system genes, and thus may have contributed to disease in this infant. *Pseudomonas aeruginosa* was only detected after antibiotic treatment, and also has many predicted type III and toxin genes, as well as type VI secretion system genes (*Supplementary file 4*). Other potentially significant bacteria were strains of *C. sporogenes* and *Paenibacillus*. Interestingly, the communities in infant #3 included *Bifidobacterium* (MSTE12-related), an organism that is often considered to be a beneficial commensal and not frequently observed in premature infants (*Butel et al., 2007*).

Infant #8 developed NEC 1 day after collection of the last sample. The communities in the two pairs of samples from different times on the same day (*Figure 8—figure supplement 3*) contain generally similar organisms, but rapid abundance shifts occur, consistent with general observations over whole day periods. Especially prominent in samples from infant #8 were a *Klebsiella pneumoniae*-related strain (*Figure 4—figure supplement 2*) and an *E. cloacae* (*Figure 4—figure supplement 1*) strain. *C. perfringens* has a notable inventory of predicted pathogenicity-related genes. *E. coli*, present in the three samples collected prior to diagnosis, may have contributed to intestinal inflammation, given that it has a large inventory of type III and type VI secretion system genes and many toxin-encoding genes (*Supplementary file 4*).

Samples from infants #9 and #10 (both of whom developed NEC) were collected only after diagnosis. Notable in the post-treatment communities from infant #9 were *E. faecalis*, *Candida parapsilosis* (see below), and some *Staphylococcus* and *Streptococcus*. A variety of Lactobacilli and *E. coli* were prominent in infant #10.

In infants who developed NEC, the prominence of many potentially pathogenic organisms is striking. Although our results do not suggest that a single organism (abundant or not) caused NEC in the studied infants, bacteria that may have contributed to NEC were present. Infants who were not diagnosed with NEC were likewise colonized by a wide variety of potentially pathogenic bacteria and some strains were even shared by NEC cases and controls. If gut colonization by pathogenic bacteria is a significant factor in the development of NEC, other health and/or environmental attributes may ultimately determine which infants become sick. Due to the small number of cases studied to date and the large number of potentially important variables, a reliable model that predicts NEC development without over-fitting cannot be constructed at this point. However, accumulation of additional data may enable the construction of such a model in the future.

## Potential roles for plasmids, phage, and bacterial–phage interaction

The gastrointestinal tract can host a complex mixture of mobile elements, including phage, viruses, plasmids, and conjugative transposons, that can transfer virulence and antibiotic resistance factors (*Salyers et al., 2004*; *Schjørring et al., 2008*; *Minot et al., 2011*). To consider the possibility that a mobile element, moving around the NICU (potentially independently of the host bacterium), was the common factor leading to NEC, we compared all sequences from all samples that were binned as plasmid-like, phage or phage-like, or of unknown origin. We commonly found essentially identical

sequences in different samples from the same infant and a few identical plasmids and phage were detected in different infants (e.g. one complete, circular plasmid from infant #6 that is affiliated with a *Clostridium* species, based on sequence similarity (*Supplementary file 3*), also occurs in infants #5 and #8). However, no mobile elements were shared by all sick infants, or by all infants diagnosed with NEC (see 'Materials and methods' section).

We leveraged the fact that some bacteria have CRISPR loci to determine whether bacteria colonizing the gastrointestinal tract of newborns have CRISPR-Cas-conferred immunity to co-occurring phage. This is important because phage sensitivity could explain rapid shifts in organism abundance. Our analyses focused on *E. faecalis* because it was abundant and widely distributed over the infant cohort. In no case did we identify an *E. faecalis* CRISPR spacer with a perfect match to any phage that coexisted in the same community (*Supplementary file 6*). However, we detected imperfect matches between *E. faecalis* CRISPR spacers and phage in the same sample, and some spacers matched perfectly to phage in other samples from the same infant and to phage in another infant. One spacer in the CRISPR locus of *E. faecalis* from infant #3 targets a prophage integrated into the genome of *E. faecalis* from infant #5 (*Supplementary file 6*). These observations indicate recent exposure of *E. faecalis* to phage populations related to those that coexisted in the NICU during the study period and suggest phage sensitivity of bacterial populations in this early gut colonization period.

Notable in samples 2 and 3 from infant #1 were Enterobacteriales phage, the genomes of which were 95× and 30× more abundant than the genome of the probable *Klebsiella* host (*Supplementary file 3*). Interestingly, ddPCR shows that the period of phage proliferation corresponded to an increase in overall cell numbers by a factor of 10 over ~5 days (*Figure 1*). *K. oxytoca* must account for this increase in bacterial cell numbers because it is essentially the only species in early-sampled communities. We infer that phage predation moderated the *Klebsiella* bloom and probably facilitated the subsequent establishment of the more complex community in later-collected samples.

## Do the NEC incident statistics support the existence of an infecting pathogen?

Our data provided a unique opportunity to study at high resolution possible factors that may have been responsible for the cluster of diagnosed NEC cases in the summer of 2014. We were able to eliminate the possibility that a single bacterial strain was the causative agent in all cases, and also did not find any support for a causative role of specific mobile elements, or particular metabolic functions. In light of these findings, we turned to statistical characterization of the disease cluster. Surprisingly, despite the frequent reference to disease outbreaks in the literature, the statistical significance of disease clusters is rarely studied (*Turcios-Ruiz et al., 2008*; *Meinzen-Derr et al., 2009*). Here, we analyzed 67 months of monthly counts of NEC diagnoses in the NICU of Magee-Womens Hospital to determine whether the apparent outbreak in the summer of 2014 was a statistically significant anomaly (*Figure 9A*). Monthly statistics are collected for other purposes and are based on different criteria for NEC, as outlined by the Vermont Oxford Network (VON). Infant #3, and another infant not enrolled in our study, were excluded due to lack of pneumatosis or pneumoperitoneum on X-rays.

No seasonal or otherwise periodic patterns in NEC diagnoses were observed, and no correlation between the number of NEC cases and daily average of hospitalized infants in the NICU was detected. Data were modeled using Poisson and negative binomial (NB) distributions and maximum likelihood estimates of the corresponding parameters were extracted (Poisson: $\lambda = 1.90$, NB: $r = 5.81$, $p = 0.75$). Data were somewhat over-dispersed relative to the Poisson distribution (*Figure 9B*) and fit the negative binomial distribution modestly better ($R^2 = 0.85$ for Poisson model, $R^2 = 0.95$ for NB model), in line with a potential dependency between diagnosed cases.

While the eight cases from summer 2014 that met VON criteria are undoubtedly at the high end of the scale, they could not be established as statistically significant, assuming these underlying Poisson or NB distributions. Inclusion of additional sick infants who do not meet the VON criteria could change the conclusion, but unfortunately monthly statistics for all diagnosed cases were unavailable. Results were essentially unchanged when data were normalized to the average daily NICU occupancy. Future study of clusters of NEC events should be tested for their statistical significance to evaluate whether consideration of a single infective agent is appropriate.

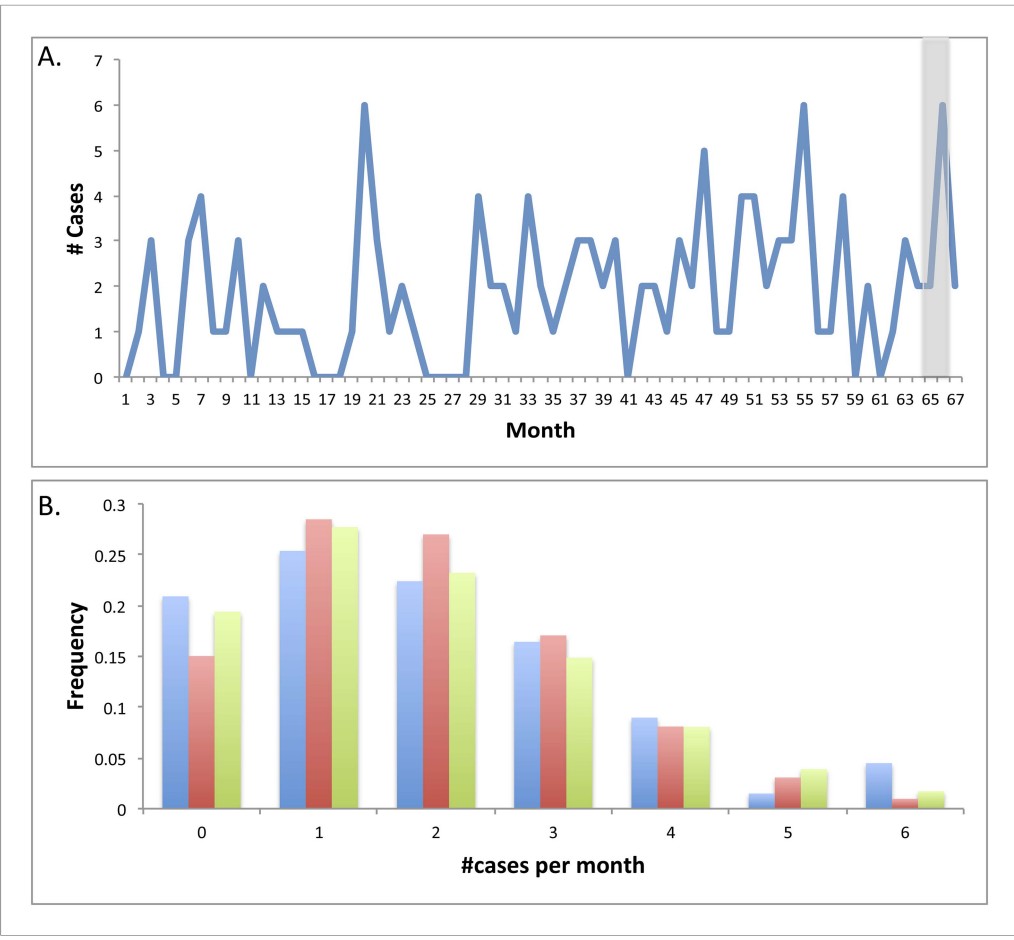

**Figure 9**. Statistical evaluation of the clustering of necrotizing enterocolitis cases during 2009–2014. (**A**) The number of diagnosed necrotizing enterocolitis (NEC) cases meeting the stringent Vermont Oxford Network (VON) criteria over 67 months. Gray shading highlights the studied period. (**B**) Observed frequency of each value of monthly NEC cases in collected data (blue); expected frequency from a Poisson (red) and negative binomial (NB; green) distributions that were fit to the observed data using maximum likelihood parameter estimation (Poisson: λ = 1.90, NB: r = 5.81, p = 0.75).

## Genome recovery from metagenomic datasets

The approach used here can, in a cost- and time-effective manner, generate very good draft genomes. For example, two near-complete *Veillonella* genomes (>4 Mbp) were assembled into 12 and 15 pieces, four *E. faecalis* genomes were reconstructed into 27–43 pieces, two *Actinomyces* into 21 and 24 pieces, one *Negativicoccus succinicivorans* genome into 12 pieces, and one *Citrobacter* strain genome into 23 pieces. Notably, some genomes reconstructed in this study represent organisms with no closely related sequenced relatives. For example, we achieved many near-complete genomes for bacteria related to *Tissierella* sp. LBN 295 (for which only four partial gene sequences are available in NCBI). This bacterium is more distantly related to, but currently profiled as related to, *Clostridium ultunense*. We also reconstructed a draft genome for an organism that we infer is related to *Peptococcus niger*. Both the *Tissierella* and *P. niger* genomes have been further curated (see 'Materials and methods' section). Also, we reconstructed hundred of genomes that, although similar to previously sequenced genomes, are different in potentially important ways (e.g. in antibiotic resistance potential and pathogenicity factors).

Interestingly, we reconstructed an ~12.7 Mbp draft genome of *C. parapsilosis*, a microbial eukaryote (fungus) that was highly abundant in the gut of infant #9 after treatment for NEC (*Figure 10A*). The genome shares >99% identity with the genome of the CDC317 isolate. Alignment

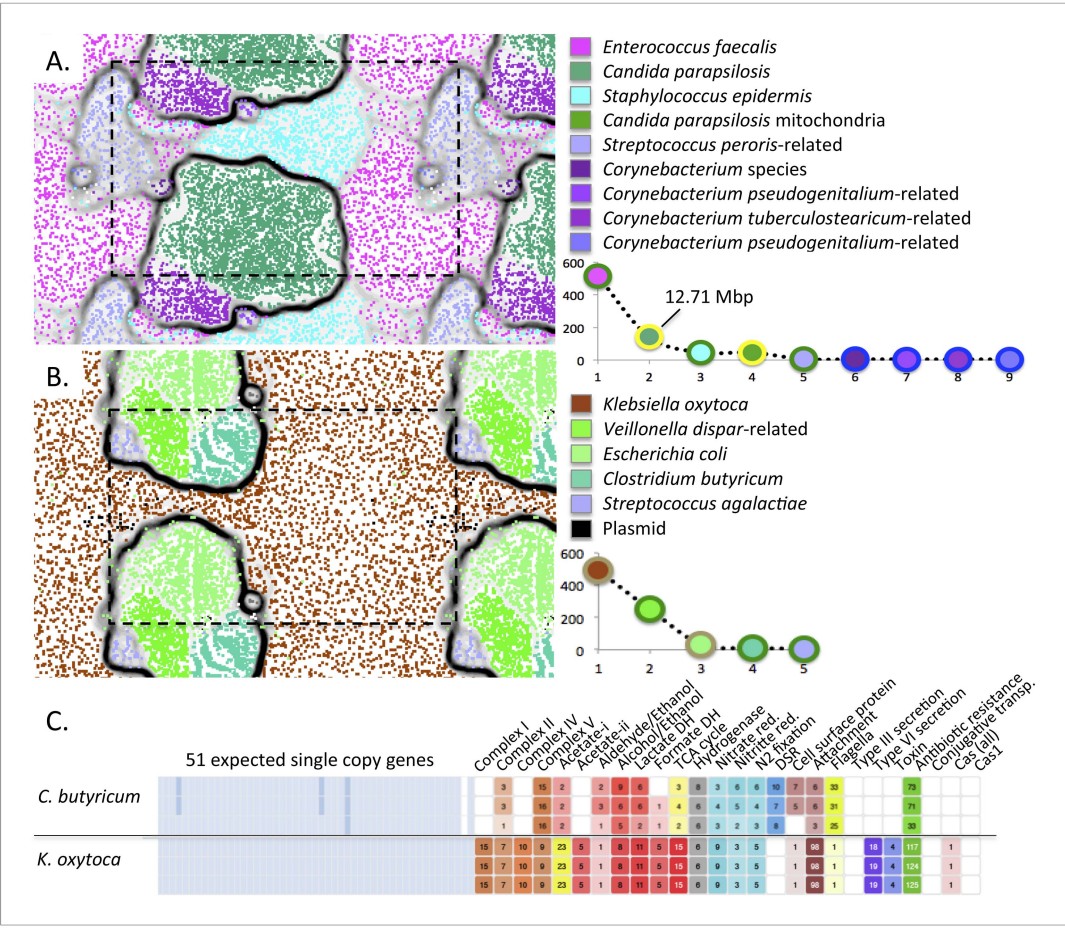

**Figure 10**. Medically important organisms were revealed by genome-resolved analyses. The emergent self organizing maps illustrate bin accuracy (dashed boxes show the periodicity of the maps) and rank abundance curves (lower right) indicate community structure. (**A**) *Candida parapsilosis* was present in infant #9 after treatment for necrotizing enterocolitis. Due to the large genome size, *Candida parapsilosis* accounts for the majority of DNA in this sample. (**B**) *Streptococcus agalactiae* (also known as group B streptococcus, GBS) was detected, albeit at low abundance, in infant #1. It is likely that the GBS caused the septic episode. (**C**) Overview of the metabolic potential for two organisms showing very different inventories of type III, VI secretion system, toxin, and antibiotic resistance genes.

The following figure supplement is available for figure 10:

**Figure supplement 1**. Mauve genome alignment (*Darling et al., 2010*) of the CDC317 *Candida parapsilosis* genome and the genome reconstructed in the current study from infant #9, sample 1 showing overall synteny and high sequence identity.

---

of our genome with the CDC317 genome verified the overall accuracy of our assembly, a notable finding given that very few microbial eukaryote genomes have been recovered previously from metagenomic data (*Figure 10—figure supplement 1*).

The ability of our approach to uncover organisms of clear medical relevance (and not detected otherwise) is also illustrated for infant #1, who developed early onset sepsis. Although present at very low abundance (~0.2%), we identified an organism whose 16S rRNA gene sequence shares 99% identity with *Streptococcus agalactiae* (group B streptococcus, GBS), in samples 4 and 5 (*Figure 10B* and *Supplementary file 3*). The infant's mother tested positive on her GBS surveillance swab and the placenta was found by pathologists to contain trace amounts of GBS. However, the newborn blood culture was negative for GBS. Likely, this organism caused the episode of early onset sepsis in this infant.

The ability to profile the genome for clinically relevant traits such as pathogenicity and antibiotic resistance is illustrated for *K. oxytoca* from infant #1 (*Figure 10C*). The low predicted pathogenicity potential of *C. butyricum* is shown for contrast (*Figure 10C* and *Supplementary file 4*).

### Genome-resolved metagenomics on a clinically relevant timescale

The time between collection of the last fecal sample and receipt of DNA sequence information from all 55 samples was 10 days. For a group of three samples, as might be collected in a clinical setting, the time required for DNA extraction is a couple of hours and for sequencing, about 2 days (the sequencing speed is method dependent). Sequence assembly, annotation, and data import into our metagenomics analysis system required 6–8 hr/sample and the time required for a first round binning that largely defined community composition and metabolic potential (generated automatically) was ~10 to ~60 min/sample. ESOM-based bin confirmation (which may not be needed for some applications) requires 1–2 hr/sample. Thus, new bioinformatics approaches tested here enable comprehensive genome-based analyses on a timescale that approaches that required for some clinical applications, for example involving patients with long-term health issues. The analysis time will decrease as sequencing technologies continue to improve and with automation of binning steps. This will make strain-resolved analysis clinically relevant for a wider range of applications, potentially including acute illnesses like NEC.

### Conclusions

We applied newly developed methods to rapidly and extensively resolve into genomes, sequence data from gastrointestinal tract-associated microbial communities from premature infants. All microbial communities in all infants sampled prior to onset of NEC harbored organisms with significant pathogenicity potential. However, we found no evidence for one common, abundant (or even minor) genomically distinct infective agent. If bacteria contribute to NEC, the effect more likely is due to exposure to a variety of potentially dangerous hospital-associated bacteria. A major finding is that the dominant population of each bacterium acquired by each infant was generally genotypically distinct. Extremely closely related organisms were only identified in a handful of cases, and occurred in both sick and healthy infants. Yet, the fact that they were identified at all, and the detection of shared minor strains of *E. faecalis* in a few cases, confirms that dispersal can occur among infants in the NICU. Overall, we suspect the existence of significant barriers that limit establishment of strains during the early stages of colonization of the premature infant gastrointestinal tract.

## Materials and methods

### Medical information

Fecal samples for enrolled infants were collected as available. From the pool of all available samples, we selected for sequencing 55 samples from which sufficient amounts of DNA were extracted. Our selection of samples from infants who did and did not develop NEC was aimed to provide dense sampling around dates leading to and following cases of NEC diagnosis in the NICU. The sampling schedule is shown in *Figure 1*, and additional medical details for all infants are provided in *Supplementary file 1*.

### Ethics statement

The study was performed with approval from the University of Pittsburgh Institutional Review Board under protocol number PRO10090089, and written parental consent was obtained on behalf of the neonates.

### Metagenomic analysis details

Sequencing reads of 160 bp in length were processed with Sickle (*Joshi and Fass, 2011*) (v1.33; available at https://github.com/najoshi/sickle) to trim both ends to remove low quality base calls. After trimming, reads were assembled with idba_ud (*Peng et al., 2012*) (v1.1.1; available at http://i.cs.hku.hk/~alse/hkubrg/projects/idba_ud/) using default settings. Resulting scaffolds >1000 bp were annotated. We used prodigal (*Hyatt et al., 2010*) (v2.60; available at https://github.com/hyattpd/Prodigal/releases/tag/v2.60) to predict genes using default

settings for metagenomics gene prediction. Protein sequences were searched against KEGG (*Kanehisa et al., 2014*) (KEGG FTP Release 2014-07-07; available at http://www.kegg.jp/kegg/download/), UniRef100 (release 2014_07; available at ftp://ftp.uniprot.org/pub/databases/uniprot/previous_releases/release-2014_07/), and UniProt (*Leinonen et al., 2004*) (same as UniRef) using USEARCH (*Edgar, 2010*) (v7.0.1001; available at http://www.drive5.com/). Additionally, reciprocal best-blast hits were determined. All matches with bit scores greater than 60 were saved, and reciprocal best hits with a bit score greater than 300 were also cataloged. We identified 16S rRNA sequences using Infernal (v1.1; available at http://infernal.janelia.org/) using default settings. The rRNA genes were predicted using Infernal (*Nawrocki and Eddy, 2013*) and tRNAs using tRNAscan_SE (*Lowe and Eddy, 1997*) (v1.23-r2; available at http://lowelab.ucsc.edu/tRNAscan-SE/). Scaffolds, gene predictions, and all associated annotations were uploaded to ggKbase.berkeley.edu for binning and analysis (http://ggkbase.berkeley.edu/project_groups/necevent_samples).

We estimated detection sensitivity for bacterial populations using the data size per sample and assuming a genome size of ~3 Mbp. For example, the sample with the least amount of data was from infant #4, sample 4 (2 Gbp). This amount of data would allow detection (4× coverage) of an organism with a 3 Mbp genome that comprised 0.6% of the sample.

Evaluation of genome completeness relied in part on the number of expected single copy genes that were identified per bin. A bin was classified as very good if the genome size was not vastly different from that of genomes of closely related organisms and most single copy genes were present in one, and only one copy. The accuracy of our genome completeness statistics was somewhat affected by genes that were split by scaffold ends. Partial genes (<50% of the gene) were not counted.

## Phylogenetic profile

As the accuracy of binning depended in part on the quality of the phylogenetic profile, we tested two approaches. First, we inventoried the best matches of proteins on each scaffold by comparison to the UniRef100 database. As this did not provide sufficient taxonomic resolution, we adopted a second approach in which the profiles were established by comparison to the much larger UniProt database. In both cases, the phylogenic classification required that ≥50% of predicted proteins on a fragment had shared affiliation at some taxonomic level. If ≥50% of predicted proteins had best matches to the same species in the database, that scaffold was profiled as that species. If ≥50% had best matches to the same genus (but not the same species), the profile assigned was of that genus. This process continued, until each scaffold had been assigned a profile at some taxonomic level. Some scaffolds were assigned the profile 'unknown' because ≤50% of predicted proteins had hits to the same domain (these scaffolds were often from viruses and plasmids).

## Binning

Binning was carried out via an online interface within ggKbase (http://ggkbase.berkeley.edu/). When using this interface, the user selects a group of genome fragments (scaffolds) based on a specific phylogenetic profile, and/or scaffold coverage and/or GC information. The amount of sequence information, the number of expected single copy genes, the number of ribosomal proteins, and bin coverage statistics are displayed for the selection. For human microbiome research, usually the choice of scaffolds is first based on phylogenetic profile and is then fine-tuned by selection of a specific subset of scaffolds based on their coverage and/or GC content. If the bin size and single copy gene inventory are appropriate, the group of scaffolds is then binned. Following one round of binning (10–60 min/sample), typically ~1 Mb of sequence information per sample was left unassigned to any organism or phage/plasmid.

Typically, the identity of genomically sampled organisms was determined based on overall sequence similarity to previously known genomes. In many cases we reconstructed partial or complete 16S rRNA genes and used this sequence information to inform organism classifications, although the presence of these genes in multiple copies often resulted in misbinning of small scaffolds encoding this gene (see notes in *Supplementary file 3*). An advantage of the presence of multiple copies of the 16S rRNA gene per genome is that it can allow us to detect populations that are otherwise at such low abundance that they would be invisible based on their overall genome coverage. The 16S rRNA scaffolds were the only parts of some very low abundance genomes detected for this reason.

The correctness of the assignment of scaffolds to genomes was verified with emergent self organizing maps (ESOMs), a clustering tool (*Ultsch and Moerchen, 2005*) that was applied to scaffold tetranucleotide frequency information (*Dick et al., 2009*). In most cases, data points assigned to the same bin clustered into clearly defined and generally strongly bounded regions of ESOMs, supporting the accuracy of the binning method. Some bin adjustments were made based on the ESOM analyses.

When the approach described above was insufficient to resolve bins for closely related species/ strains (e.g. Enterobacteriales in infant #8), we constructed ESOMs using patterns of abundance of the organisms over the time series of samples from an infant (*Sharon et al., 2013*), in combination with GC content. This led from minor to substantial improvements in bin purity and completeness.

Up to eight near-complete genomes (≥94% of expected single copy genes identified) were reconstructed per sample, and 221 near-complete genomes were reconstructed over the study. This accounting under-represents the completeness of the analysis because the presence of multiple highly related Enterobacteriales genotypes in many samples resulted in partial and fragmented recovery of specific conserved ribosomal proteins. When including Enterobacteriales genomes of the expected size but lacking these specific ribosomal proteins, 260 near-complete genomes were reconstructed.

Rank abundance curves were constructed based on coverage. For this analysis, coverage values were normalized to account for differences in data size per sample.

## Comparative genomic analysis

Strain comparison was done for genomes with >0.5 Mbp of recovered sequences, and was mostly based on alignment of sequences for 51 predicted single copy genes, many of which were ribosomal proteins. For cases with inconclusive results, mostly due to highly fragmented or very partial genomes in which many of these genes were missing, entire genome bins were aligned. In a few cases, mostly when verification of very small differences was required, manual curation of results based on inspection of read mapping to the regions in question was performed to detect local mis-assemblies.

### Alignment of single copy genes

To avoid detection of false differences based on local scaffolding errors, predicted single copy genes with one or more base pairs that were not covered by at least one perfectly matching read, or, in which >50% of mapped reads did not agree with the assembled sequence, were removed from analysis. Split genes were also removed from analysis, to avoid errors introduced at the scaffolding step.

Pairs of genomes with the same species assignment, and with at least 20 single copy genes that passed filtering were compared to each other by aligning the assembled gene sequences using nucmer (*Delcher et al., 2002*). Genome pairs that shared at least five single copy genes that passed filtering and for which all shared single copy genes were identical across their length were considered to be the same strain.

### Alignment of genome bins

Genome bins were aligned using nucmer (*Delcher et al., 2002*). The number of base pairs in alignments with over 98% identity was tallied (a higher identity threshold was not used in order to take into account occasional local scaffolding errors). If over 90% of bin length (for the shorter genome bin) was aligned using this threshold, the genomes were considered indistinguishable.

### Strain comparison via read mapping

For *E. faecalis* (*Figure 6*) and *C. paraputrificum* (*Figure 6—figure supplement 1*), strains were compared by mapping reads from different samples to specific scaffolds. Multiple alignment of the consensus sequences resulting from each mapping provided a detailed view of SNPs and indel regions while avoiding false differences resulting from partial assemblies or from assembly and scaffolding errors. Mapping was done using bowtie2 and multiple alignment was done using default parameters for the MAFFT algorithm (*Katoh et al., 2002*) implemented in Geneious v7.1.7 (*Kearse et al., 2012*).

### Detailed whole genome comparisons

In a few cases (comparison of *E. faecalis* genomes in infants #3 and #5, and in infants #2 and #7 as well as comparison of *C. parapsilosis* to the CDC317 isolate), a more detailed whole genome comparison was performed in order to locate sequence regions that were not shared between strains.

Mauve genome alignment software (*Darling et al., 2010*) was used to perform pairwise comparisons of strains of the same species assembled from different babies. Genomes were first ordered relative to a reference genome from the same species (*E. faecalis* 62, gi 323478858; *C. parapsilosis* CDC317, gi 218176216) and then compared to each other. Stretches of DNA in one of the genomes that could not be aligned to the other genome (termed 'islands') were extracted. Stringent post-processing steps were taken to filter out islands that could have resulted from missing or problematic segments in the assembly rather than actual differences in genomic sequence. Islands whose sequence was dominated by Ns (assembly gaps) were removed from further analysis. Islands that were very close (<100 bp) to scaffold edges or islands whose flanking regions mapped to two different scaffolds in the other genome, could have resulted from missing assemblies, and were therefore disregarded. To verify suspected islands, reads from samples of both babies were mapped to the island sequence and manually inspected.

### Single copy gene and CRISPR locus analyses

Geneious software v7.1.7 (*Kearse et al., 2012*) was used to align individual single copy gene sequences and for manual curation of the CRISPR loci. We used the online CRISPR spacer and repeat finder tools to recover spacer and repeat sequences (http://crispr.u-psud.fr).

### Comparison of phage, plasmid, and mobile elements

Scaffolds longer than 5000 bp that were unbinned or were binned as plasmid, phage, or mobile elements, were extracted and aligned to each other (using nucmer [*Delcher et al., 2002*]). Scaffolds that were 99% identical across 90% of their length were considered closely related.

### Genome completeness and metabolic profiling

An overview of the metabolic potential associated with genomes reconstructed in this study was established by searching the functional predictions for specific annotation terms. The number of genes that have the selected annotation terms is displayed in a table format in which rows are genomes and columns list the number of genes in each category (see *Supplementary file 4*). The search and exclusion terms for each functional category can be found via the ggKbase list function.

### Genome curation methods and results

Genomes of *C. parapsilosis* (infant #9), a species related to *C. ultunense* (infant #3), a Clostridiales from infant #6, a *V. parvula*-related strain (infant #3), an *Actinomyces* species (infant #4), and a *N. succinicivorans* strain (infant #5) were chosen for curation because they were significant and/or of very good draft quality. The curation used programs for correcting mis-assemblies and improving assemblies (Sharon et al., in preparation), which were identified through read mapping as follows. First, all reads were mapped to the genomes using bowtie2 (http://www.nature.com/nmeth/journal/v9/n4/full/nmeth.1923.html) with the –sensitive option. Next, short deletions (which we found to be common in idba-ud assemblies) in the assembled sequences were identified based on the read mappings. Last, all regions on the genomes with exceptionally low coverage were checked by collecting reads that map to those regions and their mate pairs and re-assembling them. Improvement of assemblies was achieved through read-mapping based identification of scaffolds that could be elongated or connected to other scaffolds. Both elongations and connections were achieved through local assembly of reads that were mapped to the analyzed regions and their mate pairs. For the *Candida* genome, our pipeline corrected 106 mis-assemblies (about one mis-assembly for every 120 Kbp) and reduced the number of scaffolds from 401 to 348.

### Droplet digital PCR methods

To quantify bacterial load in infant fecal samples, ddPCR was performed on the Bio-Rad QX200 platform using EvaGreen-based chemistry (Bio-Rad, Hercules, CA). A conserved, approximately 150 bp region flanking the V7 region of the 16S rRNA gene was targeted, as it has been successfully used in other probe-based qPCR assays in the past (1048f: GTGSTGCAYGGYYGTCGTCA, 1194r: ACGTCRTCCMCNCCTTCCTC [*Ramirez-Farias et al., 2009*; *Kennedy et al., 2014*]). Sample gDNA was diluted to 1:1000 and used as template in a PCR reaction consisting of 0.25 µl of 10 µM forward and reverse primer, 12.5 µl of 2× ddPCR EvaGreen Supermix (Bio-Rad), and 12 µl of template, totaling 25 µl. This PCR mix was used to create droplets following the manufacture's instructions.

Thermocycling parameters were: (1) 95℃ for 10 min, (2) 95℃ for 30 s, (3) 61℃ for 30 s, (4) 72℃ for 30 s, (5) 40 cycles (go to steps 2–4 ×39), (6) 98℃ for 10 min, and (7) 12℃ forever. All ramp rates were at 2.5℃/s. Each reaction was done in triplicate. Analysis of the ddPCR data was conducted with the QuantaSoft software package (Bio-Rad) and negative/positive thresholds set manually (just above the negative population). To calculate cell density, the copies/µl output from QuantaSoft was normalized by grams of fecal mass used for each gDNA extraction reaction. To broadly correct for copy number, the assumption of four copies per bacteria was used (*Hospodsky et al., 2012*).

## Data dissemination

The sequence information can be accessed via NCBI, accession # SRP052967. All metagenomic data associated with this study can be accessed via the ggKbase NECEvent project: http://ggkbase. berkeley.edu/project_groups/necevent_samples.

Note that this is a 'live data' repository, so that errors found after publication will be corrected and more highly curated assemblies may be available. A snapshot of the published dataset is also available for download.

## Acknowledgements

Funding was provided by NIH grant 5R01AI092531, a Sloan Foundation grant APSF-2012-10-05, and DOE Kbase grants DE-SC0004918 and ER65561. TRS received a long-term EMBO fellowship. We thank Alvaro Gonzalo Hernandez of the Roy J Carver Biotechnology Center, University of Illinois, for rapid sequencing turnaround, Wayne Getz and Alan Hubbard for helpful discussions about statistical analyses, and Elizabeth Costello for comments on the manuscript.

## Additional information

### Funding

| Funder | Grant reference number | Author |
|--------|------------------------|--------|
| National Institutes of Health (NIH) | 5R01AI092531 | Tali Raveh-Sadka, Brian C Thomas, Brian Firek, Itai Sharon, Robyn Baker, Michael J Morowitz, Jillian F Banfield |
| Alfred P. Sloan Foundation | APSF-2012-10-05 | Brandon Brooks, Jillian F Banfield |
| U.S. Department of Energy | DE-SC0004918 | Tali Raveh-Sadka, Brian C Thomas, Andrea Singh, Cindy J Castelle, Jillian F Banfield |
| U.S. Department of Energy | ER65561 | Tali Raveh-Sadka, Brian C Thomas, Andrea Singh, Cindy J Castelle, Jillian F Banfield |
| EMBO | Long term fellowship | Tali Raveh-Sadka |

The funders had no role in study design, data collection and interpretation, or the decision to submit the work for publication.

### Author contributions

TR-S, JFB, Wrote the manuscript, Conception and design, Analysis and interpretation of data; BCT, Designed the data analysis platform, Assembled and annotated data, Developed analysis scripts, Contributed to article revision; AS, Implemented and innovated the binning tools, Contributed to article revision; BF, BB, RB, MG, Contributed to article revision, Acquisition of data; CJC, Contributed to article revision, Analysis and interpretation of data; IS, Developed analysis scripts, Contributed to article revision; MJM, Contributed to article revision, Conception and design, Acquisition of data

### Author ORCIDs

Brandon Brooks, [iD] http://orcid.org/0000-0002-3738-1383

## Ethics

Human subjects: The study was performed with approval from the University of Pittsburgh Institutional Review Board under protocol number PRO10090089, and written parental consent was obtained on behalf of the neonates.

# Additional files

### Supplementary files

• Supplementary file 1. Clinical characteristics of infants in this study. Necrotizing enterocolitis (NEC) was defined as definite NEC (Bell's stage II or III). CS: caesarean section; V: vaginal delivery; BM: breast milk; combination indicates a combination of breast milk and infant formula.

• Supplementary file 2. Overview of samples, the day of life on which each sample was collected, the original internal database sample number, the amount of data that was generated per sample after trimming to remove low quality bases, the amount of data that went into genome bins, percentage of all data that went into assemblies that ended up in bins, the amount of data that remained unbinned, the number of genome bins per sample, and the number of features (genes) per sample. Red lines indicate timing of necrotizing enterocolitis diagnosis. For two infants, samples were only available after diagnosis.

• Supplementary file 3. Overview of the bins from each sample and each infant. UNK indicates a bin of unclassified sequences. Coloring of the bin names generally corresponds to colors used in emergent self organizing map and rank abundance curves in the figures. SCG is the number of single copy genes identified per bin out of 51 expected genes.

• Supplementary file 4. Overview of predicted metabolic potential and genome completeness indicators for all moderately to well sampled genomes from all infants. Note that the single copy gene inventory underestimates genome completeness for Gammaproteobacteria when multiple species were present (see main text). Each line represents a genome bin. The bin name provides information about the sample of origin: the first digit is the infant number, the second is the sample number (e.g. 1_2 is the second sample from infant #1) and the organism type. Lists to the right profile the electron transport chain in which the presence of a terminal oxidase, in combination with a TCA cycle, indicate aerobic metabolism. Information about pathways involved in fermentation processes, nitrogen metabolism, the cell surface and secretion, motility, toxicity and pathogenicity, mobile elements, and CRISPR-based virus defense and antibiotic resistance is also shown. Note that *Peptoclostridium difficile* and *Clostridium difficile* are equivalent.

• Supplementary file 5. Analysis of deeply sampled *Enterococcus faecalis* populations to detect sequencing reads with SNPs consistent with their derivation from populations present in other infants. Less deeply sampled populations typically had no SNPs consistent with derivation from a population present in another infant.

• Supplementary file 6. In no case does *Enterococcus faecalis* have a spacer that can silence a phage present in the same community. Green text highlights cases where mutations likely prevent silencing.

### Major dataset

The following dataset was generated:

| Author(s) | Year | Dataset title | Dataset ID and/or URL | Database, license, and accessibility information |
|---|---|---|---|---|
| Raveh-Sadka T, Thomas BC, Singh A, Firek B, Brooks B, Castelle CJ, Sharon I, Baker R, Good M, Morowitz MJ, Banfield JF | 2015 | Sequence Information | SRP052967 | Publicly available at NCBI Sequence Read Archive (http://www.ncbi.nlm.nih.gov/sra). |

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
