## [Decision Letter]

Thank you for sending your work entitled “Gut bacteria are rarely shared by
co-hospitalized premature infants, regardless of necrotizing enterocolitis
development” for consideration at *eLife*. Your article has been
favorably evaluated by Detlef Weigel (Senior editor), Roberto Kolter (Reviewing editor),
and two reviewers.

The Reviewing editor and the two reviewers discussed their comments before we reached
this decision, and the Reviewing editor has assembled the following comments to help you
prepare a revised submission.

You present your work detailing the microbiota of the feces in premature infants with
and without necrotizing enterocolitis (NEC) using metagenomic methods. Overall we found
that the findings presented are a very important contribution to the field. While the
fact that the authors did not find of any commonality of microorganisms that contribute
to NEC is somewhat disappointing, the finding of the unique microbiota at a strain level
for most infants in this study will shape the way scientists and physicians will think
about disease pathogenesis and gut colonization in these infants.

The manuscript presents a large amount of important work, but they are currently
presented as a loose compilation of findings. As it stands, one comes away feeling that
less has been accomplished than is actually the case. Thus, the authors should make
every effort to revise the written presentation so as to generate a text that flows
better and makes a compelling case for the importance of the findings, despite the
current lack of a common thread leading to NEC. In revising the text, the authors should
pay particular attention to responding the following major concerns, some of which ask
for analyses that may indeed provide interesting insights:

1) What were the strain level differences between closely related taxa? The suggested
reasons for the high level of unique diversity between infants are compelling, but we
would like to understand how different these strains are? Are we talking major
rearrangements, or SNP variance? Even with partial to near-complete fragments the
authors should be able to start to think about this. This would help to narrow down
which of the several possibilities provided by the team are likely. Do any of the
differences seem likely to have occurred due to local niche differentiation as a result
of host genetics? Might it be possible to generate a figure that shows, perhaps by
alignment, the degree of similarity between the different strains of the same species
between different infant guts?

2) While there is no particular suite of taxa that are always present with NEC, is it
possible that Random Forest could be used to determine the predictive potential of a
subset of genomic strains for NEC, or other health statistics that were also
collected?

3) Figure 1 is not well explained in the
manuscript. While it is useful to see an actual timeline with the dates of the samples,
the authors don't really make a correlation between bacteria per gram of feces
and anything else in the manuscript. Also, there should be some text in the manuscript
detailing the time points sampled for these infants.

4) The authors’ method is presented as a two-week protocol and authors claim that
this is 'clinically relevant time scale.' However, in the case of NEC,
12-14 days is in fact not clinically relevant. Neither the time frame nor the amount of
labor involved in producing these types of results can be considered clinically
relevant. There is the possibility of the current analyses to be eventually modified to
become useful in clinical practice, but that would require considerable modifications.
The authors do not provide any description as to how the protocols would have to be
modified to actually make them clinically relevant, as two weeks for a diagnostic test
for NEC would be of no utility in the diagnosis or treatment of these patients. Thus,
the authors should revise the manuscript to indicate that their current protocols are in
fact not really clinically relevant and that modifications would have to be developed to
be useful in diagnostics.

---

## [Author Response]

*1) What were the strain level differences between closely related taxa? The
suggested reasons for the high level of unique diversity between infants are
compelling, but we would like to understand how different these strains are? Are we
talking major rearrangements, or SNP variance? Even with partial to near-complete
fragments the authors should be able to start to think about this. This would help to
narrow down which of the several possibilities provided by the team are
likely*. *Do any of the differences seem likely to have occurred due
to local niche differentiation as a result of host genetics? Might it be possible to
generate a figure that shows, perhaps by alignment, the degree of similarity between
the different strains of the same species between different infant guts?*

We thank the reviewers for this important comment. We have added a new figure and figure
supplement (Figure 6, Figure 6—figure supplement 1) that provide a visualization
of differences between strains for *Enterococcus faecalis* (a highly
prevalent species) and *Clostridium paraputrificum* (one of the rare
species in which one genotype was shared among some of the infants). Differences were
visualized by mapping reads from various samples from which different strains were
recovered to a specific scaffold from one of these assemblies, and constructing a
multiple alignment of the consensus sequences derived from each mapping. In the case of
*E. faecalis* we selected a 1 Mbp scaffold recovered from infant 9 and
constituting a third of the genome. This view shows that most differences can be
attributed to SNP and small indels, but that those are present throughout the sequence
(∼90% identity for most strain pairs in the multiple alignment), suggesting that
most sequence pairs did not diverge very recently. A few larger indel regions (20-30
Kbp) were detected and contained among other things a sucrose metabolism operon, mobile
elements and genes related to Fe-S protein biogenesis. However, some strains
(specifically two pairs of strains, the one in infants #3 and early samples of
infant #5, and the one in infants #2 and #7) were much more similar
to each other, with very few detected differences across the compared sequence. Judging
by the small extent of differences and their type (e.g., a prophage insertion in infants
#2 and #7), it is possible that differentiation between those closely
related strains occurred very recently, perhaps even within the host, though not
necessarily due to host genetics.

*2) While there is no particular suite of taxa that are always present with NEC,
is it possible that Random Forest could be used to determine the predictive potential
of a subset of genomic strains for NEC*, *or other health statistics
that were also collected?*

While a predictive model is definitely a holy grail in the study of NEC, given the small
number of studied cases so far and the large number of potentially contributing factors,
a reliable model that predicts NEC development without over-fitting cannot be
constructed at this point. However, accumulation of similar data may ultimately enable
the construction of such models. We are in the process of acquiring additional data, and
hope that we will be able to present such models in the future. Presently however, we
would like to note that current data does not show any strong correlation between NEC
and any of the health statistics collected. We now discuss this issue in the main
text.

*3)*
Figure 1
*is not well explained in the manuscript. While it is useful to see an actual
timeline with the dates of the samples, the authors don't really make a
correlation between bacteria per gram of feces and anything else in the manuscript.
Also, there should be some text in the manuscript detailing the time points sampled
for these infants*.

We thank the reviewers for drawing our attention to this issue. Discussion of these
results was inadvertently left out. We now discuss the strategy for the sampling
schedule shown in this figure. In addition, we note that quantification of the bacterial
load in each sample was in general agreement with previous measurements in full-term
infants of similar postnatal ages (11), and that the measured variation in the number of microbes per gram
feces did not exceed a 100 fold across all samples. Finally, we state that we did not
observe a consistent trend of change in bacterial load prior to or following diagnosis
of NEC.

*4) The authors’ method is presented as a two-week protocol and authors
claim that this is 'clinically relevant time scale.' However, in the
case of NEC, 12-14 days is in fact not clinically relevant. Neither the time frame
nor the amount of labor involved in producing these types of results can be
considered clinically relevant. There is the possibility of the current analyses to
be eventually modified to become useful in clinical practice, but that would require
considerable modifications. The authors do not provide any description as to how the
protocols would have to be modified to actually make them clinically relevant, as two
weeks for a diagnostic test for NEC would be of no utility in the diagnosis or
treatment of these patients. Thus, the authors should revise the manuscript to
indicate that their current protocols are in fact not really clinically relevant and
that modifications would have to be developed to be useful in
diagnostics*.

The reviewers are correct by saying that the current timescale of our analyses is not
relevant for diagnosis and treatment of NEC cases. When we referred to clinical
relevance, we meant to suggest that this timescale could be relevant in other, less
acute, clinical situations where long-term health problems are involved. In the future,
as technology and algorithm improvement significantly reduce these timescales, this type
of analyses could be made possible also for real-time management of acute illnesses like
NEC. We have now clarified this in the text, and we hope that the reviewers will find
our re-phrasing of this section satisfactory.